# Outcome Comparison between Low-Dose Rabbit Anti-Thymocyte Globulin and Basiliximab in Low-Risk Living Donor Kidney Transplantation

**DOI:** 10.3390/jcm9051320

**Published:** 2020-05-02

**Authors:** Sang Jin Kim, Jinsoo Rhu, Heejin Yoo, Kyunga Kim, Kyo Won Lee, Jae Berm Park

**Affiliations:** 1Department of Surgery, Samsung Medical Center, Sungkyunkwan University School of Medicine, Seoul 06351, Korea; khjginigini@hanmail.net (S.J.K.); jinsoo.rhu@samsung.com (J.R.); 2Statistics and Data Center, Research Institute for Future Medicine, Samsung Medical Center, Seoul 06355, Korea; heejin17.yoo@sbri.co.kr (H.Y.); Kyunga.j.kim@samsung.com (K.K.)

**Keywords:** basiliximab, rabbit anti-thymocyte globulin, graft survival, kidney transplantation

## Abstract

The objective of this study was to compare outcomes between basiliximab and low-dose r-ATG in living donor kidney transplantation recipients with low immunological risk. Patients in the low-dose r-ATG group received 1.5 mg/kg of r-ATG for 3 days (total 4.5 mg/kg). Graft survival, patient survival, acute rejection, *de novo* donor specific antibody (DSA), estimated glomerular filtration rate (e-GFR) changes, and infection status were compared. Among 268 patients, 37 received r-ATG, and 231 received basiliximab. There was no noticeable difference in the graft failure rate (r-ATG vs. basiliximab: 2.7% vs. 4.8%) or rejection (51.4% vs. 45.9%). *de novo* DSA was more frequent in the r-ATG group (11.4% vs. 2.4%, *p* = 0.017). e-GFR changes did not differ noticeably between groups. Although most infections showed no noticeable differences between groups, more patients in the r-ATG group had cytomegalovirus (CMV) antigenemia and serum polyomavirus (BK virus) (73.0% vs. 51.9%, *p* = 0.032 in CMV; 37.8% vs. 15.6%, *p* = 0.002 in BK), which did not aggravate graft failure. Living donor kidney transplantation patients who received low-dose r-ATG and patients who received basiliximab showed comparable outcomes in terms of graft survival, function, and overall infections. Although CMV antigenemia, BK viremia were more frequent in the r-ATG group, those factors didn’t change the graft outcomes.

## 1. Introduction

Immunological rejection is known to increase the risk of graft loss after kidney transplantation (KT) [1]. Formerly, induction immunosuppressant agents such as basiliximab (Simulect^®^, Novartis Pharmaceuticals), an interleukin-2 receptor monoclonal antibody (IL-2 RA) and rabbit anti-thymocyte globulin (r-ATG, Thymoglobulin^®^, Sanofi) were used to reduce early acute rejection. r-ATG is known to have a higher immunosuppressive effect than basiliximab. However, it also has a higher risk of enabling infection [2]. Therefore, the relative risks of acute adverse reaction and subsequent infection are commonly compared when selecting an induction agent. Basiliximab is often used for immunologically low-risk patients, and r-ATG can be used for high-risk patients. However, the choice of induction agent and dosing is still debatable.

A randomized controlled study comparing r-ATG and basiliximab (total 278 patients) has shown a lower incidence of acute rejection and a higher incidence of infection in the r-ATG group [3]. However, the r-ATG dose used in that study was intermediate (1.5 mg/kg for 5 days). In addition, that study focused on deceased donor KT. Recently, newer efforts have been made to lower the dose of r-ATG to prevent infection. Those efforts are important, especially in low-risk recipients of living donor kidney transplantation (LDKT).

Our aim in this study is to compare graft survival and patient survival between basiliximab and low-dose r-ATG in low-risk LDKT patients in a single transplantation center. Other outcomes such as biopsy-proven acute rejection (BPAR), renal function changes, infections, and the development of *de novo* donor specific antibody (DSA) were also reviewed.

## 2. Materials and Methods

### 2.1. Patient Selection 

At Samsung Medical Center, 1306 patients received KT between 11 June 2003 and 30 April 2016. The number of LDKT recipients with either related or unrelated donors was 886. Low dose r-ATG has been used in LDKT patients at this center since 2011. Thus, we narrowed the study period to be from January 2011 to April 2016. Patients who received multi-organ transplants, patients younger than 18 years, and patients receiving re-transplants were excluded. Patients who received combined ATG and rituximab or high-dose ATG were also excluded. We defined ‘immunological low-risk’ patients to be those without pre-operative DSA who had an ABO blood type compatible with the donor’s blood type, regardless of whether the living donor was related or unrelated to the recipient. Patients with pre-operative DSA or positive cross-matching of human leukocyte antigen (HLA) were thus also excluded. This study was approved by the IRB at our institution (IRB number 2019-09-066).

### 2.2. Groups with Induction Agents

Patients were divided into two groups according to the use of induction agent: patients who received low-dose r-ATG and patients who received basiliximab. r-ATG was administered at 1.5 mg/kg intraoperatively (day #0) and on post-operative days #1 and #2 (total: 4.5 mg/kg for each patient). Basiliximab was given at 20 mg just before reperfusion and on post-operative day #4, according to the manufacturer’s instructions.

### 2.3. Baseline Characteristics

The age, sex, and pre-operative creatinine levels of the donors were collected. For recipients, age, sex, body mass index (BMI), and duration of dialysis before transplantation were collected and compared. Causes of end-stage renal disease (ESRD), such as diabetes nephropathy, hypertensive nephropathy, focal segmental glomerulosclerosis (FSGS), and IgA nephropathy, were evaluated. Other underlying diseases, including diabetes mellitus (DM), cardiovascular disease, and cerebrovascular disease, were also evaluated. To determine the immunological state, the number of HLA mismatches and the percentage of positive panel reactive antibody (PRA) tests were reviewed. The cold ischemia time (CIT) and warm ischemia time (WIT) during transplantation were also analyzed.

### 2.4. Maintenance Agent

Most patients received a calcineurin inhibitor (CNI, tacrolimus or cyclosporine), mycophenolate mofetil (MMF), and prednisone as maintenance immunotherapy. The target CNI levels were high for the first two weeks (10–12 mg/dL for tacrolimus, 200–250 ng/mL for cyclosporin) to prevent early-period rejection. Dose tapering was done gradually after the first two weeks. MMF was administered at 750 mg orally twice a day. Dose reduction or temporary cessation was done when patients had abdominal pain, diarrhea, leukopenia, or infection. Steroid pulse therapy was done with a mini-dose (500 mg of intravenous methylprednisolone before reperfusion and post-operative day #1) followed by dose tapering to 250 mg, 125 mg, and 75 mg on days #2–4, respectively, and 60 mg on days #5–7. After that, 16 mg of prednisolone was given orally twice a day for one week and then tapered to 8 mg twice a day until 4 weeks after surgery. The immunosuppression regimen and doses were modified for patients who were > 65 years because they are more susceptible to infection than younger people. For those patients, the target CNI level for the first two weeks was 8–10 mg/dL for tacrolimus or 150–200 ng/mL for cyclosporin, and MMF was administered at 500 mg orally twice a day. This dose reduction was also applied case-by-case to patients with infection. 

### 2.5. Graft Failure and Overall Survival

We compared the graft failure and patient survival of the two groups. ‘Graft failure’ was clinically diagnosed with nephrologist when patient’s kidney graft function was declined and patient needed chronic dialysis or re-transplantation due to aggravating uremia. Patients who eventually suffered from acute kidney injury and received hemodialysis for a short time were not considered to have had a graft failure. Patient survival during follow up was also compared between the two groups. In addition, we defined the status of a patient who needed dialysis within one week after transplantation until graft function recovered as ‘delayed graft function’ [4].

### 2.6. PRA Screening and HLA Single Identification

PRA values and HLA mismatches were evaluated pre-operatively. After transplantation, we checked PRA once a week until discharge. When the result of a PRA screening was positive or whenever a patient’s creatinine level was elevated, HLA single identification was checked with a Luminex^®^ assay (Luminex Corporation). Patients who developed DSA after transplantation that was not present before the operation were said to have ‘*de novo* DSA’.

### 2.7. CMV (cytomegalovirus) Infection Screening and Management

As a prophylaxis for CMV infection, our center administered i.v. ganciclovir at 5 mg/kg daily during the hospital stay to CMV IgG negative patients when the donor was CMV IgG positive. At discharge, oral valganciclovir was prescribed for long-term prophylaxis for only 10 weeks before 2015 due to a limitation in the national insurance coverage. After 2015, insurance coverage was extended, so valganciclovir was prescribed for 200 post-operative days. We also administered i.v. ganciclovir prophylaxis to patients in the r-ATG group for two weeks. For screening, our center checked CMV antigenemia every week during the post-operative period. CMV antigenemia checks were also done at every month’s visit from discharge until post-operative 12 months and weekly for re-admitted patients. Patients with CMV antigenemia of more than 50/400,000 white blood cells (WBCs) were treated with ganciclovir (intravenously, 5mg/kg q 12 hr) or valganciclovir (orally 900 mg bid) as preemptive therapy until the antigenemia disappeared. The doses of these agents were reduced when patients’ creatinine clearance decreased. In addition to the antiviral agent, MMF was removed and the CNI dose was reduced during recovery from a CMV infection. Treatment duration was extended for patients who suffered from CMV disease, such as CMV esophagitis or colitis, until they had fully recovered.

### 2.8. BK Polyomavirus Infection Screening and Management

Our center used an algorithm to detect and treat BK virus (Figure 1), based on a previous study at our center [5]. Urine BK virus polymerase chain reaction (PCR) detection was checked at post-operative #1, 5, 9, 16, 24, 36, and 48 weeks and whenever a patient’s creatinine level showed elevation. If the result was positive, urine BK virus PCR quantitation was performed. If urine BK virus was more than four log copies/mL, MMF was reduced or stopped as a preemptive therapy. Serum BK virus PCR quantitation was done if the urine BK virus was more than seven log copies/mL. When the serum BK viral load was more than four log copies/mL or BK nephritis was suspected clinically, a renal biopsy was done. If BK nephritis was diagnosed pathologically or BK viremia was persistent, CNI was replaced with an mammalian target of rapamycin (m-TOR) inhibitor (usually sirolimus).

### 2.9. Other Infections

Our center used i.v. ceftizoxime (1 g q 12 hr) as prophylaxis against peri-operative bacterial infection until post-operative day #2 (in case the r-ATG group, day #5). For prophylaxis against fungal infections such as *Pneumocystis jirovecii*, one tablet of Bactrim™ was given daily for 6 months beginning on post-operative day #5. We also checked patients admitted to the hospital due to an infection and those who had infections during admissions for other problems. Simple infections, such as herpes zoster in a localized area or an upper respiratory infection that could be controlled in an outpatient clinic, were not counted in this study. The types of infection were categorized as viral pneumonia, bacterial or fungal infections, and tuberculosis. Most viral infections, except CMV and BK, are simple and do not cause persistent harm to patients, so only viral pneumonia was analyzed in our study.

### 2.10. Biopsy Proven Acute Rejection

When a patient’s creatinine became elevated without the appearance of other problems that can influence kidney function such as fever, dehydration, or vascular problems, a renal biopsy was done to detect possible rejection. Also, our center adopted protocol biopsy at 2 weeks and around 1 year, separately for KT recipients since August 2012. Renal biopsies were evaluated according to the Banff classification. Our center define biopsy proven acute rejection (BPAR) as borderline change, acute T-cell-mediated rejection (TCMR) and antibody-mediated rejection (AMR), which was separately counted and analyzed in our study. Patients with clinically-driven rejection received steroid pulse therapy (500mg of i.v. methylprednisolone for three days followed by dose tapering of 250 mg, 125 mg, 75mg and 60mg, respectively). For patients with subclinical TCMR or borderline change which was incidentally diagnosed by 2-weeks or 1-year protocol biopsy, steroid pulse therapy was done case by case considering patient’s condition.

### 2.11. Statistical Analysis

The SAS-9.4 statistical program (SAS Institute, Cary, NC, USA) was used for all statistical analyses. Differences between the two groups were analyzed with the Wilcoxon rank sum test and Fisher’s exact test. Survival outcomes and associated risk factors were tested using a Cox proportional-hazards regression analysis. Biopsy proven acute rejection, CMV, BK virus, and other infections were analyzed by logistic regression analysis. The independent variables were the recipient’s age, sex, BMI, DM status, dialysis duration, and pre-operative PRA; the donor’s age, sex, and creatinine; pre-operative HLA 1,2 mismatch; and the CIT and WIT. For risk evaluation, selection of variable for multivariate analysis was dependent on the *p*-value < 0.1 at result of univariate analysis. If there was less than two variables which satisfied the condition, alternative criteria was applied (*p*-value < 0.4 and HR > 1.29 or < 0.77) considering clinical importance [6]. The induction agent variable was also included in multivariate regardless of *p*-value. Repeated e-GFR outcomes were evaluated using a generalized estimating equation. 

## 3. Results

### 3.1. Baseline Characteristics

A total of 268 patients were included, of whom 265, 259, and 156 patients had data from 1-year, 2-year, and 5-year post-operative follow-up appointments, respectively. Of them, 37 patients received low-dose ATG, and 231 patients received basiliximab (Table 1). 

The median recipient age was 47 years. The median donor age was 44 years. The most common cause of ESRD was glomerulonephritis, such as IgA nephropathy or FSGS, followed by DM nephropathy. The recipients’ age, BMI, dialysis duration, sex, and DM status and the donor’s age and pre-operative creatinine levels were similar between the two groups. The last pre-transplant HLA mismatch and PRA values also showed no noticeable difference between groups. The WIT and CIT were slightly longer in the r-ATG group (*p*-value < 0.05), though WIT and CIT were not checked in 22 patients and 17 patients, respectively. 

### 3.2. Graft and Patient Survival

The mean follow-up duration of total patients was 5.25 years (5.27 years in basiliximab, 4.79 years in r-ATG group). The median follow-up duration was 5.33 years (5.52 years in basiliximab, 4.85 years in r-ATG group). Total 12 patients were lost to follow-up more than 1 year from data gathering time (3 in r-ATG, 9 in basiliximab group). The overall 5-year survival of the kidney graft was 96.4% (Figure 2). A total of 12 (4.5%) patients had graft failure, including one patient (2.7%) in the r-ATG group and 11 (4.8%) patients in the basiliximab group (Table 2). Five-year graft survival was 97.1% in the ATG group and 96.4% in the basiliximab group, without noticeable difference between the two groups (*p* = 0.735). The recipient age, BMI, sex, DM and donor age, sex and presence of CMV or BK virus infection had no statistical effect on the risk of graft failure (Table 3) in univariate analysis. Induction therapy, DM, HLA 2 mismatch also showed no statistical effect on multivariate analysis. Event of total BPAR showed higher risk in graft failure (HR = 5.89, *p* = 0.025) in multivariate analysis. The overall 5-year patient survival was 98.8% and did not differ noticeably between the two groups. Two patients in the basiliximab group and no patients in the r-ATG group underwent delayed graft function, but that difference was also not noticeable (*p* = 0.898).

### 3.3. Biopsy Proven Acute Rejection (BPAR) and De Novo DSA

A total of 159 patients received subclinical protocol biopsy either 2-weeks (157 patients) or 1-year (98 patients). A total of 125 (46.6%) patients had at least one event BPAR either acute TCMR or borderline change with 19 in the r-ATG group (51.4%) and 106 in the basiliximab group (45.9%), without showing noticeable difference (*p* = 0.335, Table 2.). Patients was classified into five subgroups according to BPAR status. Clinical TCMR, clinical borderline change (patients who only had clinical borderline change without clinical TCMR), subclinical TCMR (patients with TCMR at protocol biopsy and without clinical BPAR), subclinical borderline change (patients with borderline change at protocol biopsy and without any other type of rejection) and no rejection.

The rate of clinical BPAR was 30.6% (19.8% of clinical TCMR, 10.8% of borderline change) in all patients. For patients with subclinical TCMR or subclinical borderline change, steroid pulse therapy was done with 89.5% and 70% rate, respectively A graph of rejection-free survival is presented in Figure 3A–C for each total BPAR, clinical BPAR and subclinical BPAR, respectively. In all three graph, rejection-free survival did not differ between the r-ATG group and the basiliximab group. HLA 2 mismatch aggravated total BPAR in both the univariate and multivariate analyses (Table 4). HLA 1 mismatch was associated with acute rejection in the univariate analysis but not the multivariate analysis. Total four patients got AMR and all of these patients had AMR and TCMR at the same time. Two patients had spontaneous AMR and clinical TCMR in the basiliximab group. Each one patient had spontaneous AMR and subclinical TCMR in the r-ATG and the basiliximab group.

Twenty-three patients were not examined for the exact state of *de novo* DSA. Among the remaining patients (*n* = 245), five (14.3%) patients in the r-ATG group and five (2.4%) patients in the basiliximab group had *de novo* DSA (*p* = 0.004). The median time for first detection of *de novo* DSA was 26.7 months (36.6 months in the basiliximab, 14.9 months in the r-ATG group). The mean median fluorescence intensity (MFI) values were 1,926, 1,592, and 2,261 among all patients and the r-ATG and basiliximab groups, respectively, which was not differ noticeably (*p* = 0.465). In Addition to induction agent, HLA 2 mismatch was associated with a higher risk in both univariate (HR = 2.91, *p* = 0.038) and multivariate analysis (HR = 2.72, *p* = 0.046). Other factors did not show noticeably different.

### 3.4. Renal Function after KT

The mean e-GFR levels at 1 month, 1 year, 3 years, and 5 years after transplantation were 68.5, 62.2, 65.9, and 64.7 mL/min/1.73 m^2^, respectively (Table 2, Figure 4). The mean e-GFR level showed a tendency to decrease 1 year after transplantation compared with 1 month after transplantation. It then recovered at 2 to 3 years after transplantation. The e-GFR changes did not differ statistically between the r-ATG and basiliximab groups (*p* = 0.159). Younger donors, and those with longer dialysis duration before transplantation were associated with a high e-GRF level in both the univariate and multivariate analyses (Table 5). 

### 3.5. CMV, BK Virus, and Other Infections

Twenty-seven (73.0%) patients in the r-ATG group and 120 (51.9%) patients in the basiliximab group had CMV antigenemia at least once (Table 6), which was not noticeably different between the two groups (*p* = 0.032, multivariate). Old donor age was a risk factor for CMV infection in the multivariate analysis (OR: 1.024, *p* = 0.026). The number of patients with CMV antigenemia of more than 50/400,000 WBCs, who thus needed to be treated with ganciclovir, was 6 (16.2%) in the r-ATG group and 15 (6.5%) in the basiliximab group (*p* = 0.049, univariate). This event happened within 1 year after transplantation in all patients. Six patients without CMV IgG received a kidney from a CMV IgG (+) donor. Of them, five patients with basiliximab induction, who all received a short duration of valganciclovir prophylaxis (10 weeks post-operative), had CMV antigenemia post-operatively (3 patients got CMV antigenemia after valganciclovir cessation). The remaining patient received r-ATG induction and valganciclovir for 200 days and did not get CMV antigenemia.

BK uremia was detected in 18 (48.6%) patients in the r-ATG group and 112 (48.5%) patients in the basiliximab group, not a noticeable difference between the two groups (*p* = 0.985). However, BK viremia occurred more often in the r-ATG group in both the univariate and multivariate analyses (37.8% in the r-ATG group vs. 15.6% in the basiliximab group, *p* = 0.002, multivariate). Old donor age elevated the risk of BK viremia in the univariate analysis (OR: 1.031, *p* = 0.025).

No patient in the r-ATG group and seven (3.0%) patients in the basiliximab group got viral pneumonia, but that difference was not noticeable. Bacterial, fungal, and tuberculosis infections also did not differ noticeably between the r-ATG and basiliximab groups. Overall, females tended to have more bacterial infections than males (OR: 3.07, *p* < 0.001).

### 3.6. Comparison of de novo DSA in Case of CMV, BK Viral Infection

Due to high proportion of de novo DSA in the r-ATG group, sub-groups with CMV and BK virus were analyzed. In the r-ATG group, two of six patients (33.3%) with CMV > 50/400,000 showed de novo DSA, and three of 29 patients (10.3%) without CMV > 50/400,000 showed de novo DSA, for an OR of 4.333 (*p* = 0.166). Also in the r-ATG group, three of 13 patients (23.1%) with serum BK virus showed de novo DSA, and two of 22 patients (9.1%) without serum BK virus showed de novo DSA, for an OR of 3.000 (*p* = 0.268). These results were similar to the result of overall patient who were tested for *de novo* DSA (OR = 3.206, 4.732 and *p* = 0.160, 0.018 in CMV > 50/400,000 and blood BK, respectively). 

## 4. Discussion

Our study found no noticeable differences in the graft survival, five-year patient survival, acute rejection rate, or renal function between the low dose r-ATG group and the basiliximab group. However, *de novo* DSA, CMV antigenemia, and BK viremia occurred more often in the r-ATG group. Other infections, such as viral pneumonia, bacterial pneumonia, and fungal infections did not differ noticeably between the groups. 

In our center, we historically used r-ATG for a long time (7–14 days from the peri-operative day) at an earlier period than other centers and found a high risk of adverse effects, including infection. Therefore, we changed to an intermediate dose (1.5 mg/kg for 5 days), which we changed to a reduced dose (1.5 mg/kg for 3 days) in 2011. Brennan et al. showed that r-ATG at a total dose of 7.5 mg/kg can reduce acute rejection but increase the infection rate in high-risk patients [3]. Nafar et al. compared three r-ATG doses: 1.5 mg/kg for 3 days, 4.5 mg/kg once, and 2.0 mg/kg for 3 days [7]. The three different dose groups in that study showed no significant differences in rejection, but the infection rate was 23% in the group that received 1.5 mg/kg of r-ATG for 3 days, which was lower than in the groups that received 4.5 mg/kg of r-ATG once (33%) or 2.0 mg/kg of r-ATG for 3 days (30%). CMV antigenemia, length of stay, and the re-admission rate showed the same tendency.

Our results show no statistically noticeable difference in rejection or graft failure between the low-dose r-ATG and basiliximab groups of patients receiving low-risk LDKT. The rate of clinical BPAR was 30.6% (TCMR = 19.8%, borderline = 10.8%) in all patients. And for patients with subclinical TCMR or subclinical borderline change, steroid pulse therapy was done with 89.5% and 70% rate. Some patients did not receive steroid pulse therapy at time of subclinical BPAR case by case considering old age, poor general condition. This early-detection and treatment policy may lead to stable graft survival (5-year survival = 96.4%). However, event of BPAR at all time showed higher risk in graft failure (HR = 5.89, *p* = 0.025) in multivariate analysis. 

Some previous studies compared low-dose r-ATG with basiliximab. Laftavi et al. performed a study with 8 years of follow-up after kidney transplantation [8] and found decreased rejection and a lower mean creatinine level in the low-dose (total 3–5 mg/kg) r-ATG group at 3 and 5 years compared with the basiliximab group of low-risk LDKT recipients (7.8% vs. 35% for rejection, 1.2 mg/dL vs. 1.5 mg/dL for 3-year creatinine level). However, the 8-year graft survival rates were very high in both groups, with no significant difference between the two groups (100% in r-ATG vs. 98% in basiliximab). In that study, the 8-year graft survival from deceased donor KT was higher in the r-ATG group (86% vs. 76%). Another study of 46 immunologically low-risk LDKT recipients found a lower rejection rate in the r-ATG group (0%) than in the basiliximab group (23.8%) [9]. Borderline change in that study was also very lower in the r-ATG group (8%) compared with the basiliximab group (42.9%). e-GFR levels showed no significant difference between the two groups. CMV infection was more frequent in the r-ATG group, but it was effectively treated with ganciclovir. Those two studies defined a low immunological risk as PRA <30%. However, our center defined low-risk as negative DSA before surgery regardless of PRA because we found that in the absence of DSA, PRA did not influence the outcome after transplantation [10]. In our study population, only 6 (2.3%) patients had PRA > 50%, without a noticeable difference between the two groups.

DSA, either preexisting or *de novo*, is a well-known risk factor for antibody-mediated rejection of a transplanted kidney and graft failure [11]. One study of 315 KT patients showed poorer graft survival with *de novo* DSA than without *de novo* DSA (10-year graft survival: 57% vs. 96%, *p* < 0.0001) [12]. That study also showed more peritubular capillaritis in the 6-month renal biopsy in the *de novo* DSA group, even in patients with stable renal function. Our study showed that r-ATG was associated with a higher appearance of *de novo* DSA, although the number of patients with *de novo* DSA was very small (4 patients, 11.4%). A previous study has shown that r-ATG seems to be able to lower the rate of *de novo* DSA and antibody-mediated rejection compared with other induction therapies [13]. However, the r-ATG group has consistently shown a high immunological risk in comparison studies, which leads to more aggressive use of preemptive therapies such as plasmapheresis or intravenous immunoglobulin (IVIG) injection than is used in the other induction groups, including the basiliximab group. Previous studies had different circumstances from our study, which we limited to patients with low immunological risk who did not receive plasmapheresis or IVIG. One possible reason for the high rate of *de novo* DSA in the r-ATG group in our study is the high rate of CMV and BK viral infections in the r-ATG group, which led to a reduction in the maintenance immunosuppressive agent. Our center usually stopped MMF when CMV antigenemia was more than 50/400,000 WBCs or the serum BK was positive. A CNI reduction or change from CNI to an mTOR inhibitor was also applied. Additional subgroup analysis showed tendency of high proportion for *de novo* DSA in patients with CMV antigen > 50/400,000 WBCs or serum BK viral infection. These results were noticeably different in BK viremia (OR = 4.732, *p* = 0.018), but not in CMV > 50/400,000 (*p* = 0.160). The relationship between r-ATG and *de novo* DSA in low-risk patients needs to be clarified in further studies.

Our study showed no noticeable difference in the infection rate between the two groups except for CMV and BK viral infections. This result was similar to the results of other studies [3,14,15]. CMV antigenemia was frequent in all patients (54.9%), the r-ATG group (73%), and the basiliximab group (51.9%). However, high CMV antigenemia (WBC > 50/400,000) that required treatment was much lower: 16.2% in the r-ATG group and 6.5% in the basiliximab group. All the CMV infections in our study were easily controlled with i.v. ganciclovir or oral valganciclovir, and they did not influence graft failure because they were detected early. 

Our center checked serum CMV pp65 antigen levels for CMV infection screening. For monitoring CMV infections, quantitative PCR is helpful and widely used. Also, CMV antigenemia has some limitations, such as a lack of assay standardization and result interpretation and diminished performance with low absolute neutrophil count (<1000/mm^3^) [16]. One study of 217 samples from KT patients showed that PCR was slightly more accurate than antigenemia [17]. However, another study of 797 samples found good correlation between CMV antigenemia and a positive PCR result (χ^2^ = 78.05; *p* < 0.0001) [18], and a study of 899 samples showed that CMV antigenemia and PCR had similar diagnostic value [19]. Our center also previously demonstrated good correlation between CMV pp65 antigenemia (cut off value > 50/4 × 10^5^ WBCs) and PCR (DNA copies >86 copies/μL) [20]. Setting a cut off for preemptive therapy at CMV antigenemia of more than 50/4 × 10^5^ WBCs kept CMV antigenemia well controlled, without significant differences in CMV disease or graft or patient survival [21]. Although CMV antigenemia is a useful test that our center has used in KT recipients since those studies, we are planning use PCR to check CMV DNA because many recent studies have shown its effectiveness.

Among the six CMV IgG (-) recipients who received a KT from CMV IgG (+) donors, five received valganciclovir for only 10 weeks post-operatively due to a limitation of the national insurance coverage. Those five patients all got CMV antigenemia, and three of them needed preemptive therapy because their CMV antigen levels were more than 50/400,000 WBCs. On the contrary, the sixth patient, who was in the r-ATG group, received valganciclovir for 200 days post-operatively and did not get CMV antigenemia. However, the small number of cases tested here limits the statistical meaning of the results. After 2015, national insurance coverage for prophylactic valganciclovir was expanded, so our center now uses valganciclovir prophylaxis for 200 days post-operatively for all CMV IgG(+) donor to CMV IgG(-) recipient transplants, which was shown to lower the infection risk and acute rejection in a previous randomized controlled study [22].

The BK virus can be a significant risk factor for kidney graft dysfunction. It has no known specific treatment or prophylaxis [23]. Over-immunosuppression and being male or older are risk factors for BK viral infection. Moreover, ATG and a combination of tacrolimus and MMF can aggravate the infection risk [24]. Our study showed higher BK viremia in the r-ATG group than in the basiliximab group (37.8% vs. 15.6%, *p* = 0.002). To ensure early detection of BK viral infections and prevent BK nephritis, our center routinely checks urine BK virus after transplantation. If the viral amount is high, serum BK virus PCR is done, followed by a renal graft biopsy if BK nephritis is suspected. Although there is no specific treatment, we lowered the immunosuppression of patients with urine BK viral loads of more than 4 log copies/ml by switching tacrolimus to sirolimus and ceasing MMF to help patients naturally eradicate the virus. BK infection did not aggravate graft failure. A recent meta-analysis study of Scurt FG et al [25] showed patients who received ABO-incompatible KT showed higher CMV and BK viral infection rates compared to patients with ABO-compatible KT which may due to stronger immunosuppressive agent for desensitization of ABO-incompatible cases (OR = 1.27, 1.59 in CMV and BK virus, respectively). This result support the higher infection rate of CMV and BK viral infection in the r-ATG group of our study.

r-ATG induction did not aggravate bacterial infections or viral pneumonia in ways that led to patient hospitalization compared with basiliximab induction. Life-threatening infections, such as fungal or tuberculosis infections, did not differ noticeably between the groups either. This could be due to the use of Bactrim as a prophylaxis for fungal infections. Bacterial infections were more common in females regardless of induction agent because the most common bacterial infections after KT were urinary tract infections, which are more common in females generally [26,27] probably due to the short urethra and close position of the urethral opening to the anus and vagina. A previous mentioned meta-analysis of Scurt FG showed no statistical differences of UTI and PJP infections between ABO-incompatible and compatible KT group, somewhat different from our study in bacterial era. However, some of studies compared in this meta-analysis showed high rate of UTI infection in stronger immunosuppressed group. 

Choosing a proper induction agent in kidney transplantation is very important and still being researched. IL-2 receptor blockers such as basiliximab have been well studied, found to reduce rejection compared to no induction, and are used generally [28,29]. Basiliximab is replaced by r-ATG in some circumstances. Although r-ATG is well-known to reduce acute rejection in immunologically high-risk patients [3,30], it has the potential to aggravate infection risk. In immunologically low-risk patients, a study showed that r-ATG offered no benefit for rejection [31]. However, another study showed that r-ATG offered benefits in rejection and graft survival compared with an IL-2 receptor blocker [32]. Other research has shown that r-ATG has a supportive effect during steroid withdrawal after KT [33], and a recent study showed that low-dose r-ATG had a more stable effect in the steroid withdrawal group [34], though yet another study showed no superiority of r-ATG compared with basiliximab [35]. Although the debate goes on, induction with low-dose r-ATG could lower the infection risk and confer a benefit for steroid withdrawal with stable outcomes, but that possibility requires further study. In this regard, our present results showing that low-dose r-ATG and basiliximab have similar effects and complications in low-risk LDKT recipients (except for CMV and BK viral infections, which were well controlled) offers a foothold for future plans. 

One limitation of our study is the small number of patients in the r-ATG group (*n* = 37). In addition, this study was retrospective. However, it is meaningful to compare the outcomes of low dose r-ATG and basiliximab in immunologically low-risk LDKT patients, which has not been studied sufficiently. One advantage of our study is that we compared infections (not only CMV, but also BK virus and other infections) in detail.

## 5. Conclusions

Immunologically low-risk LDKT recipients receiving low-dose r-ATG and those receiving basiliximab had comparable results in terms of rejection, graft function, and graft survival. Although CMV antigenemia and BK viremia were more frequent in the r-ATG group, those infections were managed well through early detection and management and did not lead to graft failure. Other infections did not differ noticeably according to induction agent. However, the potential long term, detailed influence of the induction agents on graft function should be analyzed among more patients in following studies.

## Figures and Tables

**Figure 1 jcm-09-01320-f001:**
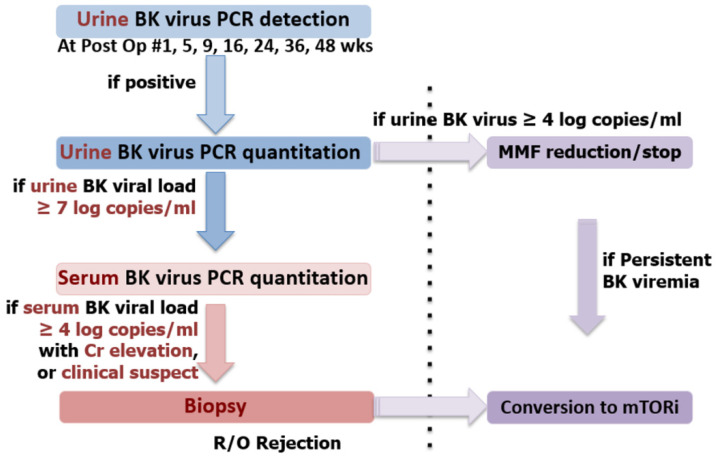
BK virus detection and treatment protocol. This serial detection method is used in our center to diagnosis the degree of BK viral infection. When the urine BK viral load is more than four log copies/mL, mycophenolate mofetil (MMF) is reduced or stopped. For BK nephritis patients, tacrolimus is replaced with an mTOR inhibitor.

**Figure 2 jcm-09-01320-f002:**
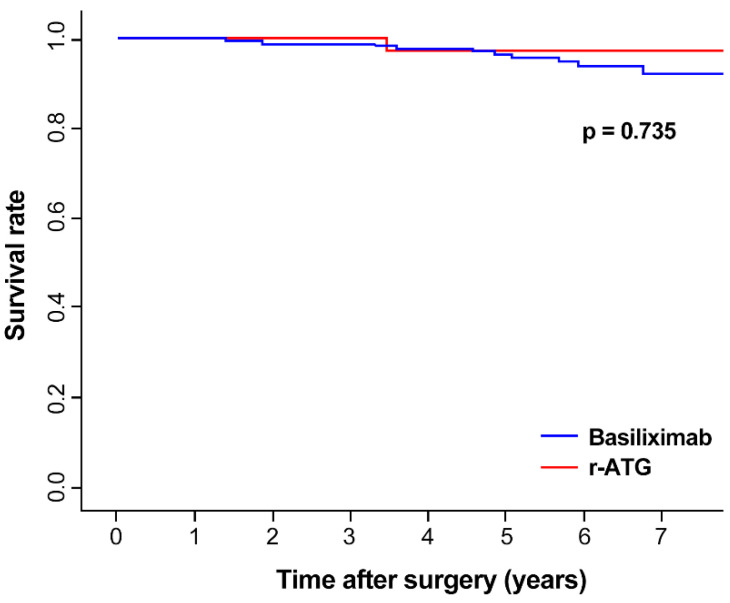
Graft survival in the r-ATG and basiliximab groups. The two groups did not differ statistically.

**Figure 3 jcm-09-01320-f003:**
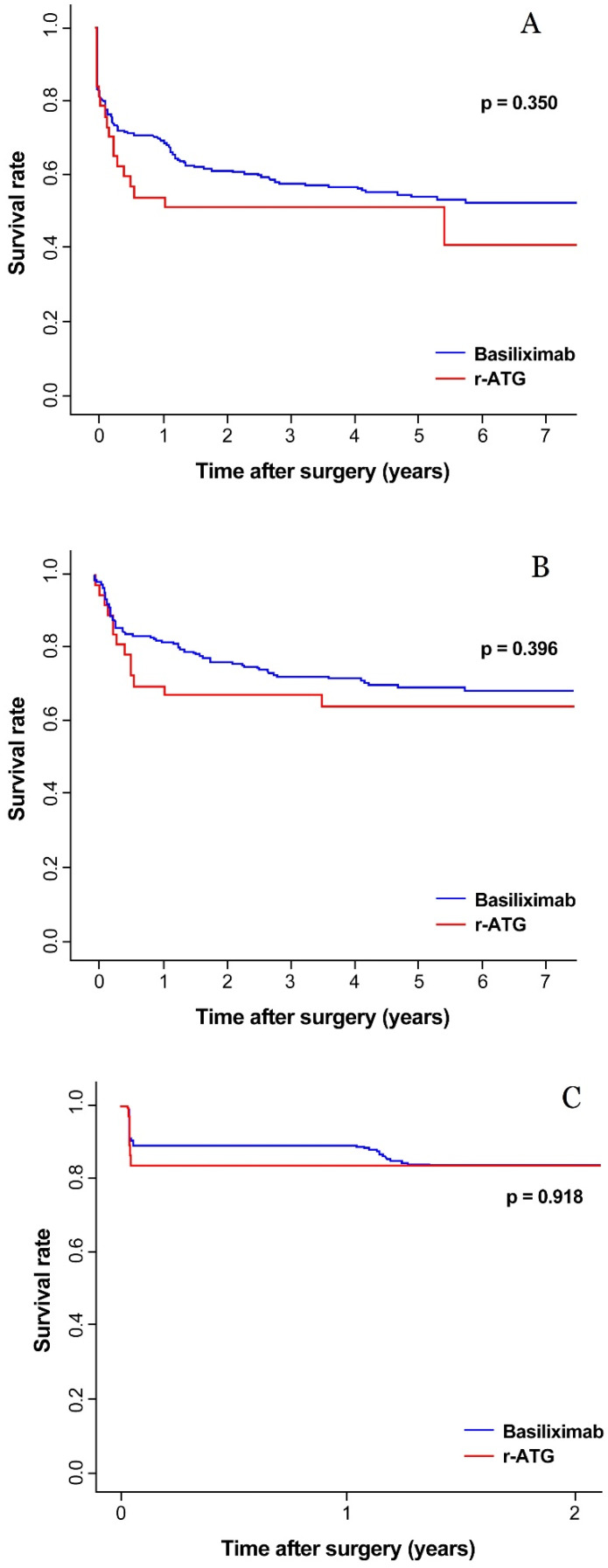
The rejection-free survival rate of the r-ATG and basiliximab in total BPAR (**A**), clinical BPAR (**B**) and subclinical BPAR (**C**).

**Figure 4 jcm-09-01320-f004:**
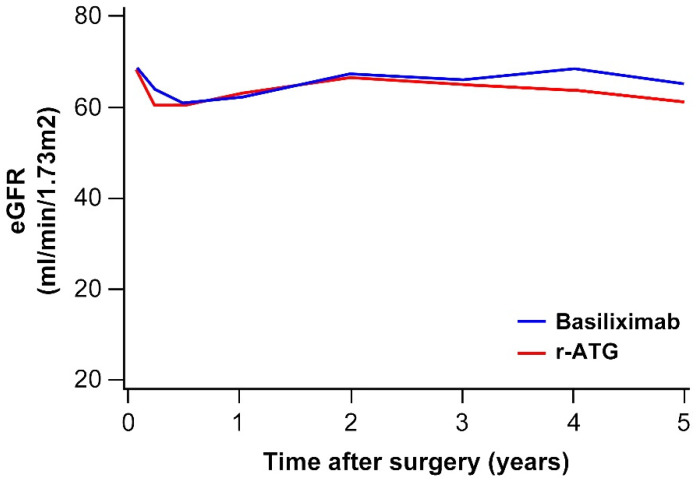
e-GFR changes in the r-ATG and basiliximab groups.

**Table 1 jcm-09-01320-t001:** Pre- and intra-operative characteristics of the r-ATG and basiliximab groups.

	Total (*n* = 268)	Low Dose r-ATG (*n* = 37)	Basiliximab (*n* = 231)	*p*-Value
Recipient				
Age (yr), median (range)	47 (19–72)	46 (19–68)	47 (20–72)	0.435
Sex (M/F)	172/96	23/14	149/82	0.783
BMI (kg/m^2^)	22.5	22.7	22.4	0.620
DM (%)	69 (25.8)	9 (24.3)	60 (22.4)	0.831
HLA Class I MM, median (range)	2 (0–4)	2 (0–4)	2 (0–4)	0.176
HLA Class II MM, median (range)	1 (0–2)	1 (0–2)	1 (0–2)	0.593
CMV status (%)				
Donor +/Recipient +	254 (94.8)	35 (94.6)	219 (94.8)	0.974
Donor +/Recipient -	6 (2.2)	1 (2.7)	5 (2.2)	
Donor -/Recipient +	8 (3.0)	1 (2.7)	7 (3.0)	
Dialysis duration (months), median (range)	7.5 (0–637)	7.6 (0–637)	7.3 (0–547)	0.981
Cause of ESRD				0.760
DM (%)	61 (22.3)	9 (24.3)	52 (22.5)	
GN (%)	88 (32.8)	14 (37.8)	74 (32.0)	
PCKD (%)	9 (3.4)	1 (2.7)	8 (3.5)	
HTN (%)	39 (14.6)	5 (13.5)	34 (14.7)	
Other (%)	14 (5.2)	3 (8.1)	11 (4.8)	
Unknown (%)	57 (21.3)	5 (13.5)	52 (22.5)	
PRA > 50%	6 (2.3)	2 (5.7)	4 (1.8)	0.182
Donor				
Age (yr), median (range)	44 (18–80)	47 (19–66)	43 (18–80)	0.095
Sex (M/F)	133/135	21/16	112/119	0.350
Cr (mg/dL), mean	0.83	0.92	0.82	0.266
WIT, mean (minutes) ^†^	3.0	3.49	2.94	0.011
CIT, mean (minutes) ^‡^	90.1	106.1	87.3	0.015

r-ATG, rabbit anti-thymocyte globulin; BMI, body mass index; DM, diabetes mellitus; HTN, hypertension; HLA, human leukocyte antigen; MM, mismatch; RRT, renal replacement therapy; GN, glomerulonephritis; PCKD, polycystic kidney disease; WIT, warm ischemia time; CIT, cold ischemia time; ^†^: WIT was not checked in 22 patients, so 246 patients were analyzed; ^‡^: CIT was not checked in 17 patients, so 251 patients were analyzed.

**Table 2 jcm-09-01320-t002:** Graft function outcomes of the r-ATG and basiliximab groups.

	Total (*n* = 268)	Low Dose r-ATG (*n* = 37)	Basiliximab (*n* = 231)	*p*-Value
DGF (%)	2 (0.7)	0	2 (0.9)	0.898
Graft failure (%)	12 (4.5)	1 (2.7)	11 (4.8)	0.737
Patient death (%)	3 (1.1)	0	3 (1.3)	0.657
Total BPAR (%)	125 (46.6)	19 (51.4)	106 (45.9)	0.335
Clinical BPAR				
TCMR (%)	53 (19.8)	4 (10.8)	49 (21.2) *	0.355
Borderline change (%)	29 (10.8)	9 (24.3)	20 (8.7)	
Subclinical BPAR ^†^				
TCMR (%)	11 (4.1)	2 (5.4) *	9 (3.9) *	
Borderline change (%)	32 (11.9)	4 (10.8)	28 (12.1)	
No rejection	143 (53.4)	18 (48.6)	125 (54.1)	
*de novo* DSA (total *n* = 245) ^‡^	10 (4.1)	5 (14.3)	5 (2.4)	0.004
e-GFR, mean (mL/min/1.73m^2^)				0.120
1 month	68.5	68.2	68.6	
1 year	62.2	63.0	62.1	
2 years	67.1	66.5	67.1	
3 years	66.0	65.9	66.0	
5 years	64.7	63.0	64.9	

r-ATG, rabbit anti-thymocyte globulin; DGF, delayed graft function; BPAR, biopsy-proven acute rejection; TCMR, T-cell mediated acuter rejection; DSA, donor specific antibody; e-GFR, estimated-glomerular filtration rate; ^†^: total 159 patients received protocol biopsy (34 and 125 in the r-ATG group and the basiliximab group, respectively); ^‡^: 35 and 210 patients were screened for *de novo* DSA in the r-ATG group and the basiliximab groups (2 and 21 patients omitted), respectively; *: Two patients had spontaneous AMR and clinical TCMR in the basiliximab group. Each one patient had spontaneous AMR and subclinical TCMR in the r-ATG and the basiliximab group.

**Table 3 jcm-09-01320-t003:** Risk analysis of graft failure.

	Univariate HR (95% CI)	*p*-Value	Multivariate ^†^ HR (95% CI)	*p*-Value
Induction therapy: r-ATG/Basiliximab	0.70 (0.09–5.49)	0.737	0.64 (0.08–5.03)	0.670
Recipient age	1.00 (0.95–1.05)	0.903		
BMI	1.08 (0.91–1.28)	0.403		
sex: male/female	1.66 (0.45–6.12)	0.450		
DM	1.86 (0.56–6.22)	0.315	1.48 (0.44–4.98)	0.525
HLA 1 mismatch	1.22 (0.74–2.00)	0.442		
HLA 2 mismatch	1.77 (0.75–4.18)	0.192	1.40(0.57–3.42)	0.462
WIT	0.95 (0.58–1.58)	0.854		
CIT	0.99 (0.97–1.01)	0.375		
Donor age	0.97 (0.93–1.02)	0.267		
creatinine	1.58 (0.22–10.0)	0.686		
sex: male/female	1.42 (0.45–4.48)	0.552		
CMV antigenemia > 50/400,000	0.04 (0.00–639)	0.523		
BKV viremia	10.91 (0.20–4.17)	0.908		
BPAR	6.72 (1.47–30.8)	0.014	5.89 (1.25–27.8)	0.025

r-ATG, rabbit anti-thymocyte globulin; BMI, body mass index; DM, diabetes mellitus; HLA, human leukocyte antigen; PRA, panel reactive antibody; CMV, cytomegalovirus; BKV, polyomavirus BK; BPAR, biopsy-proven acute rejection; ^†^: Only one variable has *p*-value lower than 0.1, so alternative criteria (*p* < 0.4 and HR > 1.29 or < 0.77) was applied for selection of variables in multivariate analysis. Three variables plus variable ‘induction therapy’ was included.

**Table 4 jcm-09-01320-t004:** Risk analysis of acute rejection and *de novo* DSA.

		Univariate HR (95% CI)	*p*-Value	Multivariate HR (95% CI)	*p*-Value
Biopsy proven Acute Rejection	Induction therapy: r-ATG/Basiliximab	1.27 (0.78–2.08)	0.335	1.16 (0.69–1.96)	0.585
	Recipient age	0.99 (0.98–1.01)	0.423		
	BMI	1.02 (0.97–1.08)	0.420		
	sex: male/female	1.39 (0.95–2.03)	0.091	1.33 (0.88–1.99)	0.172
	DM	1.29 (0.88–1.90)	0.196		
	HLA 1 mismatch	1.17 (1.00–1.37)	0.045	1.01 (0.83–1.24)	0.896
	HLA 2 mismatch	1.71 (1.31–2.23)	<0.001	1.74 (1.24–2.44)	0.001
	PRA ≥ 50%	0.28 (0.04–1.98)	0.201		
	WIT	1.12 (0.98–1.29)	0.091	1.11 (0.97–1.27)	0.149
	CIT	1.00 (1.00–1.01)	0.199		
	Donor age	1.01 (1.00–1.03)	0.105		
	creatinine	1.14 (0.52–2.51)	0.749		
	sex: male/female	0.75 (0.53–1.06)	0.104		
de novo DSA ^†^	Induction therapy: r-ATG/Basiliximab	6.83 (1.87–25.0)	0.004	6.78 (1.80–25.5)	0.005
	Recipient age	0.97 (0.93–1.03)	0.313		
	BMI	0.99 (0.81–1.21)	0.925		
	sex: male/female	0.36 (0.10–1.30)	0.119		
	DM	0.29 (0.04–2.34)	0.246		
	HLA 1 mismatch	1.24 (0.70–2.22)	0.461		
	HLA 2 mismatch	2.91 (1.06–8.00)	0.038	2.72 (1.02–7.29)	0.046
	WIT	1.06 (0.64–1.75)	0.822		
	CIT	1.08 (1.00–1.02)	0.147		
	Donor age	0.99 (0.94–1.04)	0.720		
	creatinine	2.19 (0.40–12.2)	0.369		
	sex: male/female	1.59 (0.44–5.79)	0.480		

r-ATG, rabbit anti-thymocyte globulin; BMI, body mass index; DM, diabetes mellitus; HLA, human leukocyte antigen; PRA, panel reactive antibody; WIT, warm ischemia time; CIT, cold ischemia time; DSA, donor specific antibody; ^†^: 35 and 210 patients were screened for *de novo* DSA in the r-ATG group and basiliximab groups (two and 21 patients omitted), respectively.

**Table 5 jcm-09-01320-t005:** Risk analysis of e-GFR changes.

	Univariate Beta Coefficient (Standard Error)	*p*-Value	Multivariate Beta Coefficient (Standard Error)	*p*-Value
Induction therapy (time adjusted): r-ATG/Basiliximab		0.100		0.120
Recipient age	0.05 (0.07)	0.505		
BMI	−0.45 (0.25)	0.078	−0.31 (0.20)	0.124
sex: male/female	−0.59 (1.85)	0.749		
DM	0.77 (1.83)	0.672		
Dialysis duration	1.36 (0.60)	0.036	1.20 (0.50)	0.017
HLA 1 mismatch	−0.81 (0.78)	0.302		
HLA 2 mismatch	0.14 (1.29)	0.910		
PRA ≥ 50%	−1.92 (4.53)	0.671		
Donor age	−0.58 (0.06)	<0.001	−0.57 (0.06)	<0.001
creatinine	−2.93 (3.00)	0.328		
sex: male/female	5.86 (1.70)	0.001	1.04 (1.47)	0.478

r-ATG, rabbit anti-thymocyte globulin; BMI, body mass index; DM, diabetes mellitus; HLA, human leukocyte antigen; PRA, panel reactive antibody.

**Table 6 jcm-09-01320-t006:** Infection comparison between the r-ATG and basiliximab groups.

	Total (*n* = 268)	Low Dose r-ATG (*n* = 37)	Basiliximab (*n* = 231)	*p*-Value
CMV antigenemia	147 (54.9)	27 (73.0)	120 (51.9)	0.032 ^†^
CMV antigen > 50/400,000 WBCs	21 (7.8)	6 (16.2)	15 (6.5)	0.049
BK viremia	50 (18.7)	14 (37.8)	36 (15.6)	0.002 ^†^
Viral pneumonia (%)	7 (2.6)	0	7 (3.0)	0.538
Bacterial infection (%)	72 (26.9)	12 (32.4)	61 (26.4)	0.446
Fungal infection (%)	3 (1.1)	0	3 (1.3%)	0.928
Tuberculosis infection (%)	4 (1.5)	1 (2.7)	3 (1.3)	0.524

r-ATG, rabbit anti-thymocyte globulin; CMV, cytomegalovirus; BK, polyomavirus BK; ^†^; Multivariate analysis. Other values were analyzed using the univariate method.

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
