# Peer review of "Outcome Comparison between Low-Dose Rabbit Anti-Thymocyte Globulin and Basiliximab in Low-Risk Living Donor Kidney Transplantation"

_jcm, 2020, doi:10.3390/jcm9051320_

Round 1

Reviewer 1 Report

All acute rejection episodes must be analyzed, not only T-cell or borderline rejection. T-cell and borderline histological changes must be given separately. Authors should also separately show the results of 2-week protocol and other (clinical-driven) biopsies. As any AR episode has been proven to have deleterious effect of the graft survival, so as we took together the >50% occurrence of AR and such excellent 5-year kidney graft outcomes, there is an impression that something is wrong in those data presentation or some data are unprecise.

The authors should show what was the period of mean or median follow-up time post-transplantation? What was the median time of first de novo DSA detection in both groups? What was the exact number of subjects lost to follow-up in both study subgroups?

Figure 2 shows that only 145 patients, including only 5 (five) in r-ATG group, were available 5 years after transplantation. If the lost to follow-up rate was really over 40%, I do not think that 5-year outcomes, including eGFR values and kidney graft loss, can be cited and discussed! It is a lottery, not statistics. The number of patients receiving r-ATG after 7 years in not given (zero???), but there still is a red line on the figure – why?  Besides, the definition of graft loss as a “loss the ability to produce urine” is unusual, as a majority of KTRs goes back to the dialysis program due to the worsening of uremia and rising biochemical parameters, not a loss of urine.  

Table 3: BMI has HR value of 0.40 (1.08–1.28): it is mistake here, because such results are mathematically incorrect

Authors wrote: The recipient age, donor age, presence of CMV or BK virus infection, and DM status had no effect on the risk of graft failure (Table 3). Such a conclusion could not be drawn based on the univariate, but only multivariate analysis. As the whole cohort included 268 patients (even less, because not all had DSA screening), the number of potential independent variables of 12 is too high and the results are absolutely not reliable. Thus, all multivariate analyses should be recalculated, including only 4-5 most appropriate potential explanatory variables, and the mechanism of their selection should be written in the Methods.  

The first reviewer suggested that the term “significantly” should be changed as statistical power of those findings is weak. I think that it was supposed to be done in the abstract and conclusions. Unfortunately, the authors renamed the term “statistical” to the term “noticeably” in ALL MANUSCRIPT (except the legends to the figures), which is absurd.

Reviewer 2 Report

The authors have addressed all my previous comments.

The metaanalysis of Scurt FG et al (Lancet 2019), where the differential impact of an stronger immunsuppressive therapy on viral, bacterial and fungal infections should be mentioned in text and included in the reference list.

Author Response

Response to Reviewer 2 Comments

Comments and Suggestions for Authors

The authors have addressed all my previous comments.

Point 1: The metaanalysis of Scurt FG et al (Lancet 2019), where the differential impact of an stronger immunsuppressive therapy on viral, bacterial and fungal infections should be mentioned in text and included in the reference list.

Response 1: This meta-analysis showed similar result to our study in that of CMV and BK viral infection were higher in ABO-incompatible KT cases compared to ABO-compatible cases which may due to stronger immunosuppression in ABO-incompatible patients. This support the high CMV and BK viral infection in r-ATG group of our study. This study also showed no statistical differences of UTI and PJP pneumonia infection between two groups. This is somewhat different from our study in UTI but similar in PJP. I have added the text and reference in discussion session. Thank you for your advice.

Round 2

Reviewer 1 Report

Please find the attached file with a few more comments regarding the minor corrections.

Author Response

Point 1: All those percentages should be calculated as regard to n=37 and n=231 or n=34 or n=125 biopsied patients. Now, we read that in rATG group there was only 6.7% patients without rejection!

Response 1: Thank you for precise advice. In Table 2. We changed the % of each sub-group divided by 37 in the r-ATG group, and by 231 in the basiliximab group. The reason why we divided even the subclinical BPAR group by 37 and 231 is, these percentage looks like representing the rate rejection status of each induction group better. We also found a miss-typing of ‘29’ instead of ‘20’ in clinical borderline change in the basiliximab group and corrected number. We apologize for this.

Point 2: In the response to (5), you declare that selection of variable for multivariate analysis was dependent on the p-value < 0.1 at result of univariate analysis. However, there are higher values in the table 3 –please explain this discrepancy

Response 2: We discussed about the selection of variables in case of very small number (less than two) of variables with low p-value (< 0.1) at result of univariate analysis. We decided to apply alternative criteria (p-value < 0.4 and HR > 1.29 or < 0.77 which effect size is at least more than ‘small’) considering clinical importance. And we also decided to include the ‘induction therapy’ which is the main interest of our study. Thus only one variable (BPAR) fit to 1st criteria (p-value < 0.1) in Table 3, alternative criteria was applied. Finally, three variables (recipient DM, HLA 2 mismatch, BPAR) which fit to 2nd criteria (p-value < 0.4 and HR > 1.29 or < 0.77) plus ‘induction therapy’ were analyzed by multivariate analysis. We revised this description on ‘statisctical analysis’ paragraph of method section and the end of Table 3.

This manuscript is a resubmission of an earlier submission. The following is a list of the peer review reports and author responses from that submission.

Round 1

Reviewer 1 Report

This is an interesting study comparing the benefit to risk ration of an induction therapy with ATG vs Simulect in low immunological risk living kidney donation. The results were more or less anticipated. The potency of an immunosuppressant agent reduces immunological and increases infectious complications. This issue of choosing a potent induction therapy for low immunological risk patients must be discussed in more detail.

Furthermore, following other issues should be addressed.

  1. The presentation of the main findings of the study in the abstract does not reflect their clinical implications. The statement in the conclusion part of no differences in terms of graft survival and function in both groups is in fact true but trivializes the infectious complications in the r-ATG treated patients. The higher rate of CMV and BKV infections should be mentioned in the conclusion.
  2. I wonder why that center uses ATG in low immunological risk patients.
  3. Regarding the definition of low immunological risk: what about ABO compatibility, living-related or living unrelated donors. There are no other transplantation-process related relevant data, such as warm ischemia time listed in the table with the demographics.
  4. What are with fungal infections or tuberculosis? There are both life threatening infections. The authors must explain why the avoided to analyze these infections in their patients.
  5. I cannot understand why the incidence of de novo DSA was higher in the group with the more potent induction therapy (ie r-ATG). Was there more incompliance in that group? What about the other maintenance immunosuppressive therapy? Where there any changes that lead to the production of de novo DSA or was it simply due to the reduction in IS because of the more infectious complications? Is there a sub-analysis available for the groups with infectious complications and modification of the maintenance immunosuppressive therapy?
  6. In the discussion part is mentioned that there were not significant differences in all infections but in the methods part the authors wrote that they restricted their analysis only for viral infections which they had considered to be more significant (see also point 4). There is obviously a discrepancy in the method’s and discussion’s part of the paper, which should be clarified.
  7. What kind of biopsy proven rejections occurred? (Cellular, Humoral or Mixed?). Were there any differences in the type of rejections between the groups?
  8. Why females did have more bacterial infections?

Reviewer 2 Report

The authors aimed to compare the effectiveness and safety of two different induction regimens basiliximab or low dose ATG) in low-immunological risk living donor kidney transplantation setting. However, some major methodological obstacles exist.

Generally, this study is underpowered. The low-dose ATG subgroup is disproportionally low-numbered (n=37) in comparison to basiliximab (n=231) subgroup. Thus, the majority of statistical analysis are insufficient, especially major kidney outcomes and multivariate analyses. In the case of more ATG-receiving patients, also the reported PRA and donor age between-group differences will be of statistical significance and should be included into the logistic models. Thus, the conclusions drawn by authors are not sufficiently confirmed by statistics, because not only “p” value is important in data interpretation.  

Minor comments:

Introduction:

Thymoglobulin is now produced by the Sanofi company

There are some grammar mistakes throughout all manuscript.

Methods:

What was a criterion for an induction agent choice? Was it time-dependent?

Maintenance agent: please specify how the immunosuppression drugs dosing was modified in patients >65years and those with infection?

A routine post-transplant PRA screening as a “pre-test” for a decision if Luminex anti-DSA assessment is needed is rather unusual, especially in patients with rising serum creatinine level. What was a rationale for that protocol?

Similarly, a rather short, only 10 weeks valgancyclowir treatment period used in D+R- recipients needs explanation. It is worth to notice that >50% of study patients had positive CMV antigenemia at least once. Please add the CMV donor and recipient status into the study group characteristics.

A study report very high, >50% of BPAR. Were the protocol biopsies performed and at what time point? What criteria were used for the diagnosis of AR? How authors

Statistics: please list the potential independent variables used in the logistic regression analysis.

Results:

Please add the number of patients who completed 1-, 2-, and 5-year follow-up period. How many patients had not been screened for DSA in both subgroups? It is unusual that 5-times more patients in ATG group than in basiliximab group developed de novo DSA. What were the MFI values of those post-transplant DSAs?

Table 1 – is PRA means maximum historical or last pre-transplant PRA?

If eGFR is reported, serum creatinine levels should be omitted.